# A brain-sparing diphtheria toxin for chemical genetic ablation of peripheral cell lineages

Mafalda M.A. Pereira[1,†], Inês Mahú[1], Elsa Seixas[1], Noelia Martinéz-Sánchez[2,3], Nadiya Kubasova[1], Roksana M. Pirzgalska[1], Paul Cohen[4], Marcelo O. Dietrich[5,6], Miguel López[2,3], Gonçalo J.L. Bernardes[7,8] & Ana I. Domingos[1]

Conditional expression of diphtheria toxin receptor (DTR) is widely used for tissue-specific ablation of cells. However, diphtheria toxin (DT) crosses the blood–brain barrier, which limits its utility for ablating peripheral cells using Cre drivers that are also expressed in the central nervous system (CNS). Here we report the development of a brain-sparing DT, termed BRAINSPAReDT, for tissue-specific genetic ablation of cells outside the CNS. We prevent blood–brain barrier passage of DT through PEGylation, which polarizes the molecule and increases its size. We validate BRAINSPAReDT with regional genetic sympathectomy: BRAINSPAReDT ablates peripheral but not central catecholaminergic neurons, thus avoiding the Parkinson-like phenotype associated with full dopaminergic depletion. Regional sympathectomy compromises adipose tissue thermogenesis, and renders mice susceptible to obesity. We provide a proof of principle that BRAINSPAReDT can be used for Cre/DTR tissue-specific ablation outside the brain using CNS drivers, while consolidating the link between adiposity and the sympathetic nervous system.

[1] Obesity Laboratory, Instituto Gulbenkian de Ciência, Oeiras 2780-156, Portugal. [2] NeurObesity Group, Department of Physiology, CIMUS, University of Santiago de Compostela—Instituto de Investigación Sanitaria, Santiago de Compostela (A Coruña) 15782, Spain. [3] CIBER Fisiopatologia de la Obesidad y Nutrición (CIBERobn), Santiago de Compostela 15706, Spain. [4] Laboratory of Molecular Metabolism, The Rockefeller University, New York, New York 10065, USA. [5] Section of Comparative Medicine, Yale University School of Medicine, New Haven, Connecticut 06520, USA. [6] Department of Neuroscience, Yale University School of Medicine, New Haven, Connecticut 06520, USA. [7] Department of Chemistry, University of Cambridge, Lensfield Road, Cambridge CB2 1EW, UK. [8] Instituto de Medicina Molecular, Faculdade de Medicina da Universidade de Lisboa, Lisboa 1649-028, Portugal. † Present address: Department of Neuronal Control of Metabolism, Max Planck Institute for Metabolism Research, Cologne 50931, Germany. Correspondence and requests for materials should be addressed to A.I.D. (email: dominan@igc.gulbenkian.pt).

Conditional expression of the diphtheria toxin receptor (DTR) using the Cre/lox system is a widely used tool for tissue-specific ablation of cells. However, this tool is of limited utility for targeted genetic ablation of peripheral cells because diphtheria toxin (DT) crosses the blood–brain barrier (BBB) and many peripheral Cre drivers have additional expression in the central nervous system (CNS), thus driving ablation in the brain.

The ability of a molecule to cross the BBB depends on two critical molecular properties: polarity and size. Small uncharged molecules tend to cross the BBB, whereas those that are large and/or polarized tend to have limited permeability. To limit DT action and prevent it from entering the brain, we chemically modified DT by PEGylation to develop a brain-sparing DT (BRAINSPAReDT). This chemical modification increases the size and polarity of DT and spares DTR-expressing cells in the brain. To validate this tool *in vivo*, we implemented a genetic method for regional sympathectomy using BRAINSPAReDT.

General sympathectomy has been traditionally performed with systemic injection of chemicals such as guanethidine or 6-hydroxydopamine[1]. Unlike the chemical ablation, which is systemic, surgical sympathectomy is a localized intervention that affects the sympathetic nervous system (SNS) innervation of a confined anatomical region or organ[2–4]. However, surgical sympathectomy damages nerves of unknown identity that are mixed within the SNS bundles[5]. Moreover, neither a surgical nor a chemical approach is amenable to specifically sympathectomize organs such as the adipose tissue, which covers a large body region.

Thus, genetic ablation via expression of Cre-inducible DTR and diphtheria toxin injection would be a feasible strategy to achieve a regional sympathectomy and tissue specificity[6]. Using a catecholaminergic Cre driver (*TH-Cre*), we conditionally express the DTR in the SNS and dopaminergic neurons in the brain. In these mice, BRAINSPAReDT successfully ablates peripheral catecholaminergic neurons, but not dopaminergic neurons in the brain, and thus avoids the Parkinson-like phenotype that is a consequence of dopaminergic depletion. We found that PEGylated DT (PEGyDT)-regionally sympathectomized mice have defective thermogenesis, which is a key mechanism of energy dissipation from brown and beige adipose tissues. Thus, mice regionally sympathectomized with BRAINSPAReDT are prone to obesity, despite having normal eating behaviour. These results are consistent with the role of the SNS in lipolysis and thermogenesis. We provide a proof-of-principle demonstration that BRAINSPAReDT can generally be used for Cre/DTR ablation in the periphery, thus circumventing off-target cell ablation in the brain. In particular, BRAINSPAReDT may now be used for genetic loss of function of subsets of SNS neurons. This tool should enable systematic molecular study of the SNS as well as a plethora of peripheral neurons or cells that may be labelled by CNS markers.

## Results

**Increased hydrodynamic radius and polarity of PEGylated DT.** The covalent attachment of long unstructured polyethylene glycol (PEG) chains—PEGylation—to proteins results in increased hydrodynamic radius, or size, of the biomacromolecule, as well as increased polarity[7–10]. In the case of DT, lysines were used as conjugation sites (Fig. 1a). The use of an *N*-hydroxysuccinimidyl (NHS) ester PEG derivative, NHS-PEG$_4$, enables a fast reaction with the most accessible and reactive $\varepsilon$-amine side chains of lysine on the surface of DT (Fig. 1b).

The modification state of DT was analysed by mass spectrometry to estimate how many PEG molecules were attached to DT

(Fig. 1c; Supplementary Fig. 1). DT (G$_2$–S$_{536}$) has a molecular weight of 58,336 Da, and matrix-assisted laser desorption/ionization–time-of-flight (MALDI–TOF) analysis showed the best-resolved signal as [M + 2H]$^{2+}$ peak at 29,169 $m/z$. PEGyDT showed additional peaks at $m/z$ values 29,274, 29,381, 29,484, 29,591 and 29,702 that correspond, respectively, to masses (Da) of 58,547, 58,761, 58,967, 59,181 and 59,403 (Fig. 1c). Within a margin of variation, these values are in agreement with the expected mass additions (218 Da) for PEGylation. Conversion of DT to PEGyDT was nearly complete (relative 95% conversion) and resulted in a heterogeneous mixture differing in both the copy number of PEG units and conjugation sites. On the basis of our MALDI–TOF data, each DT molecule was modified with one to five PEG units, representing an average increase of molecular weight of 436 Da.

**PEGylated DT remains fully functional after modification.** To assess the efficiency of PEGyDT in promoting cell death, fluorescence-activated cell sorting (FACS) analysis was performed using HeLa cell cultures that were pre-incubated with vehicle, DT or PEGyDT (Fig. 1d,e; Supplementary Fig. 2a,b). HeLa cells are human, and thus express DTR, which has been cloned into mice to generate the *Rosa26-LSL-DTR* mouse line[6]. As shown in Fig. 1e, in vehicle-treated cultures, after 24 h, background cell death amounted to $9.70 \pm 0.39\%$ (cell death was assessed by propidium iodide uptake). After 24 h of incubation with toxin, DT and PEGyDT (6.50 nM), the cell death was increased to $57.23 \pm 0.63\%$ and $52.57 \pm 0.04\%$ (***$P < 0.0001$, one-way analysis of variance (ANOVA), followed by Tukey test), respectively. After 48 h, the cell death was $98.57 \pm 0.10\%$ for DT and $98.60 \pm 0.05\%$ for PEGyDT, but only $7.45 \pm 0.12\%$ for vehicle-treated cells (***$P < 0.0001$, one-way ANOVA, followed by Tukey test). Hence, PEGyDT does not have any impairment in its cytotoxic activity towards DTR-expressing cells.

**BRAINSPAReDT ablates peripheral but not central neurons.** To test the tissue-specific ablation efficiency of PEGyDT in rodents, we performed histological and behavioural analysis with a catecholaminergic Cre driver line, which targets the locus of the gene encoding for tyrosine hydroxylase (TH), a major enzyme in the biosynthetic pathway of catecholamines, such as nore-pinephrine (NE) and dopamine[11]. The *TH-Cre* driver line was crossed to the *Rosa26-LSL-DTR* mouse line (*TH-Cre; LSL-DTR*), and these mice were injected intraperitoneally (i.p.) with saline, DT or PEGyDT ($0.02 \, \mathrm{pmol \, g^{-1}}$ of body weight (BW))[6]. At 24 h following injections, mice were killed to analyse the presence of the catecholaminergic marker, TH, in brain (Fig. 2a) and in nerve fibres (Fig. 2b) from subcutaneous (SubQ) white adipose tissue (WAT). As shown in Fig. 2c, the TH$^+$/4,6-diamidino-2-phenylindole (DAPI) ratio in the ventral tegmental area of the brain of mice that received DT was $4.00 \pm 1.38\%$ (***$P < 0.0001$, one-way ANOVA, followed by Tukey test), while that of PEGyDT-injected mice was $51.00 \pm 7.17\%$ (***$P < 0.0001$, one-way ANOVA, followed by Tukey test). Vehicle-injected mice had a TH$^+$/DAPI ratio in the brain of $50.18 \pm 10.22\%$. The same analysis was performed in other brain areas containing TH$^+$ neurons (Supplementary Fig. 3 for area postrema, locus coeruleus, hypothalamus and sagittal plane of substantia nigra), and the results indicate that PEGyDT does not ablate catecholaminergic neurons in the brain when injected systemically. We probed the efficiency of PEGyDT for tissue-specific ablation of sympathetic neurons in the same mice. To this end, we dissected nerve fibres in fat pads of the aforementioned *TH-Cre; LSL-DTR* mice for immunostaining against TH and $\beta_3$-Tubulin ($\beta_3$-Tub), a general marker of peripheral nerves (Fig. 2b). DT-injected mice had a TH$^+$/$\beta_3$-Tub ratio of

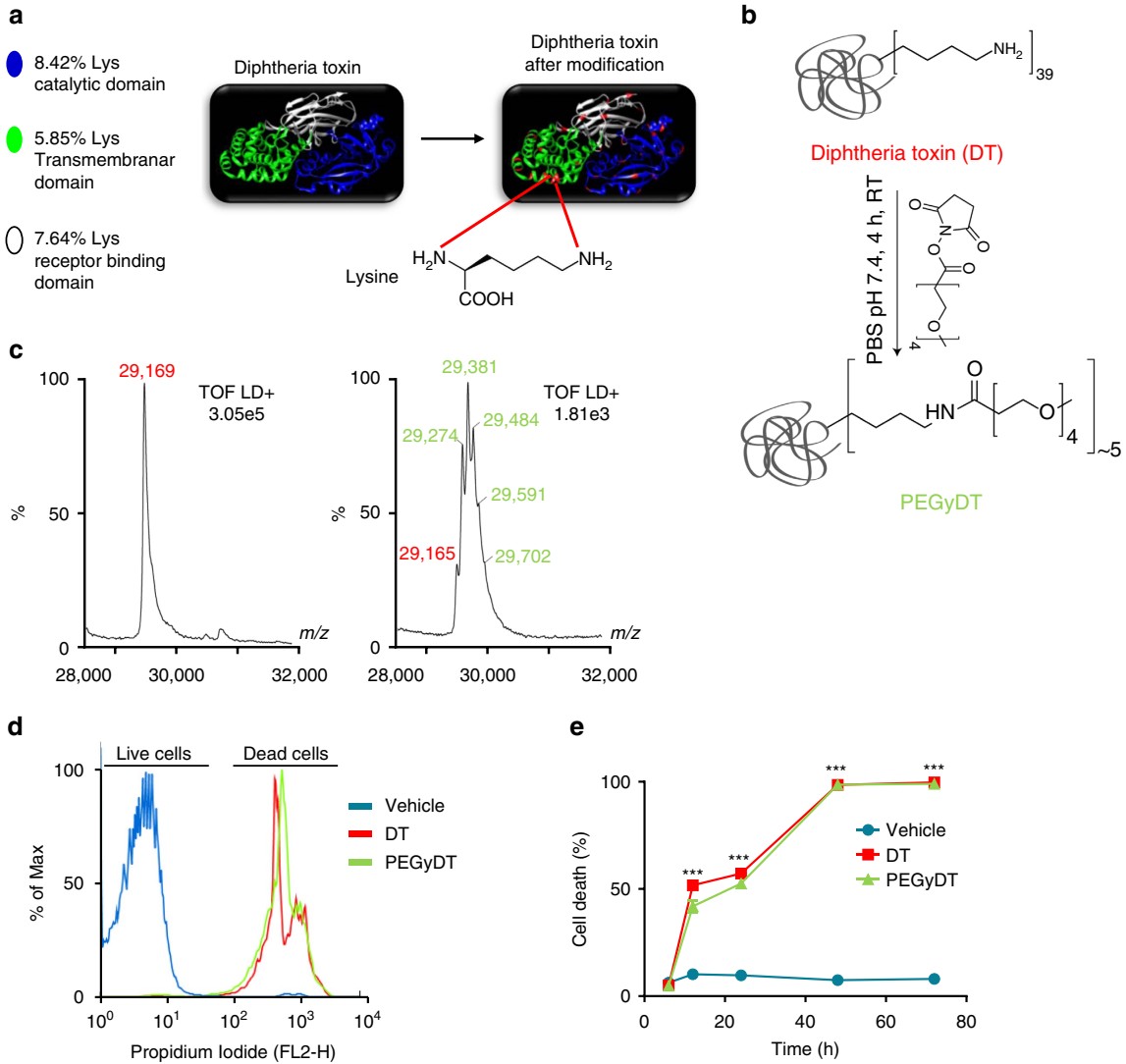

**Figure 1 | PEGylation of DT and its *in vitro* functionality after modification.** (**a**) PEG residues bind to the lysines in DT. (**b**) Schematic representation of PEGylation of DT through conjugation of NHS-PEG$_4$ at lysines. (**c**) Mass spectrometry of DT (left) and PEGyDT (right) samples. DT peak is identified in red and PEGyDT peaks are identified in green. (**d**) Live- and dead-cell populations after 48 h of incubation with vehicle, DT and PEGyDT in HeLa cells. (**e**) Time course of cell death after incubation with vehicle, DT and PEGyDT (\*\*\**P* < 0.0001, *n* = 3). Statistics were performed using one-way ANOVA test followed by Tukey test. Data are represented as mean ± s.e.m. (related to Supplementary Figs 1 and 2).

11.08 ± 1.78%, and PEGyDT-injected mice had a ratio of 11.80 ± 1.17%. These values are significantly lower than that of vehicle-injected mice in which the ratio is 102.77 ± 9.48% (\*\*\**P* < 0.0001, one-way ANOVA, followed by Tukey test), as shown in Fig. 2d. Furthermore, we confirmed that levels of TH transcripts in SNS ganglia (superior cervical ganglia) of regionally sympathectomized mice are significantly lower than that of PEGyDT-injected *LSL-DTR* mice (Supplementary Fig. 4a).

We also verified that the DT-injected mice, but not those injected with PEGyDT, had decaying health that was accompanied by severe Parkinson-like movement, weight loss and reduction in food intake that leads to premature death (Fig. 3). A blind scoring test was performed to evaluate Parkinson-like movements, which are characterized by the presence of jerks and shakes when mice initiate movement (Supplementary Movies 1 and 2). PEGyDT-injected *TH-Cre; LSL-DTR* mice moved normally, whereas DT-injected mice had marked alterations in locomotor behaviour that mimic a Parkinson-like phenotype (Fig. 3b; Supplementary Movies 1 and 2). DT-injected *TH-Cre; LSL-DTR* mice had decaying health that was evidenced by

severely reduced food intake and weight loss (Fig. 3c,e). In 1 week, the BW of DT-injected *TH-Cre; LSL-DTR* mice was reduced by 15% (Fig. 3c, \**P* < 0.01, unpaired *t*-test) and the food intake was reduced from 3 g per day to only 1 g per day (Fig. 3e, \*\**P* < 0.001, unpaired *t*-test). These decreases were not observed following injection with PEGyDT. Weight curves and food intake of PEGyDT-injected *TH-Cre; LSL-DTR* mice were similar to that of the control line *LSL-DTR* (Fig. 3d,f). The decaying health of the Parkinson-like mice required euthanasia to comply with humane criteria for animal handling. Altogether, these results indicate that PEGyDT ablates cells peripherally, but not those in the brain.

**Regional sympathectomy impairs thermogenesis and NE.** To probe a physiological effect of PEGyDT regional sympathectomy, we measured its impact on the defence of body temperature. At room temperature (RT, 25 °C), regionally sympathectomized mice had a slightly lower body temperature (36.0 ± 0.2 °C) that was not significantly different from that of PEGyDT-injected *LSL-DTR* mice, which are hereafter referred to as controls (36.4 ± 0.2 °C, Fig. 4a, *t* = 0 h, NS, *P* = 0.254, *n* = 5). As RT is only

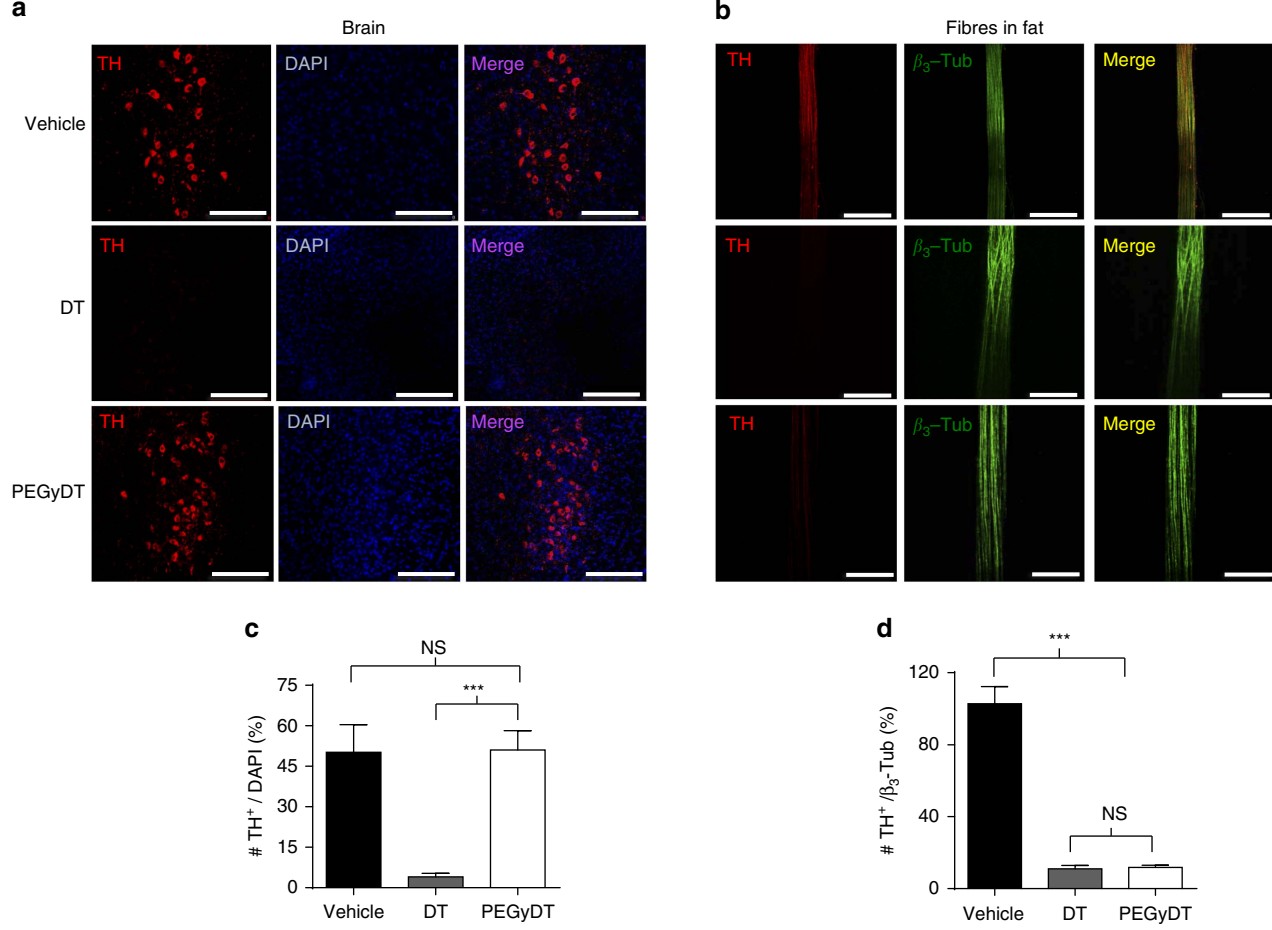

**Figure 2 | BRAINSPAReDT ablates peripheral neurons but not those in the brain.** (**a**) Confocal microscopy imaging of TH$^+$ neurons after i.p. injection of vehicle, DT or PEGyDT. Scale bar, 250 μm. (**b**) Confocal microscopy imaging of nerve fibres in fat stained for TH and for β$_3$-Tub after i.p. injection of vehicle, DT or PEGyDT. Scale bar, 100 μm. (**c**) Quantification of TH$^+$/DAPI neurons after i.p. injection of vehicle, DT or PEGyDT (***$P < 0.0001$, $n = 3$). (**d**) Quantification of TH$^+$/β$_3$-Tub neurons after *in vivo* injection of vehicle, DT or PEGyDT (***$P < 0.0001$, $n = 3$). Statistics were performed using one-way ANOVA test followed by Tukey test. Data are represented as mean ± s.e.m. (related to Supplementary Figs 3 and 4a). NS, not significant.

a few degrees below the murine thermoneutral temperature (30 °C), we challenged mice with cold (4 °C), to test the effect of regional sympathectomy on adaptive thermogenesis, which is thought to be mediated by the SNS. Control and regionally sympathectomized mice were exposed to 4 °C for 8 h, and their body temperature was measured after the challenge (Fig. 4a, $t = 8$ h). While control mice had only a 2 °C reduction in body temperature, regionally sympathectomized mice doubled that decrease, dropping to a body temperature of 32.1 ± 0.4 °C (Fig. 4a, $t = 8$ h, ***$P < 0.0001$, unpaired $t$-test). Shivering thermogenesis does seem to be majorly affected (Supplementary Movies 3 and 4). The defence of BW in response to a cold shock is typically accompanied by SNS release of NE in the brown adipose tissue (BAT). This also occurs in the SubQ WAT where a consequent increase in thermogenic gene expression in this tissue has been referred to as 'browning' or 'beiging'. However, this typical response was not seen in mice regionally sympathectomized with BRAINSPAReDT, which had significantly lower levels of NE in SubQ fat and in BAT, but not in other organs (Fig. 4b; Supplementary Fig. 4b). To explain the lack of decreased NE in some peripheral organs, we characterized *TH-Cre* by immunofluorescence on mice crossed with a reporter *LSL-ChR2-YFP* reporter. We discovered incomplete co-localization of TH and ChR2-YFP reporter in the thoracic ganglia of the sympathetic chain. We see that only 59.62 ± 4.22% of

TH$^+$ neurons are YFP$^+$ and this explains the lack of decreased NE in the organs innervated by these ganglia. However, the 13th ganglia of the sympathetic chain, which is known to innervate the adipose tissue[12], shows a co-localization of TH$^+$ neurons and GFPL10$^+$ of 94.25 ± 10.41% (Supplementary Fig. 4c). This complete overlap in fat is consistent with the marked decrease in NE content that we measure in fat (Fig. 4b; Supplementary Fig. 4b). Thus, we can conclude that, outside the brain, the *TH-Cre* driver line is a peripheral mosaic that preferentially marks neurons innervating fat, but not vital organs such as heart and lungs. The mosaicism of this *TH-Cre* driver mouse line may be due to insertional effects of the transgene, which alter the activity of the promoter in a tissue-specific manner.

Consistent with a lower NE tone in fat, we observed significantly lower levels of thermogenic gene expression in SubQ fat (Fig. 4c). These results confirm that genetic regional sympathectomy with BRAINSPAReDT has a physiological effect on adaptive thermogenesis.

To further understand the mechanism behind our data, we performed a leptin challenge (Fig. 4d). As expected the food intake was reduced to roughly 1 g per day both for controls and regionally sympathectomized mice. Nevertheless, the effects of leptin on weight loss were lower in regionally sympathectomized mice, further confirming that an intact sympathetic innervation on fat is required for leptin's effect on BW[5].

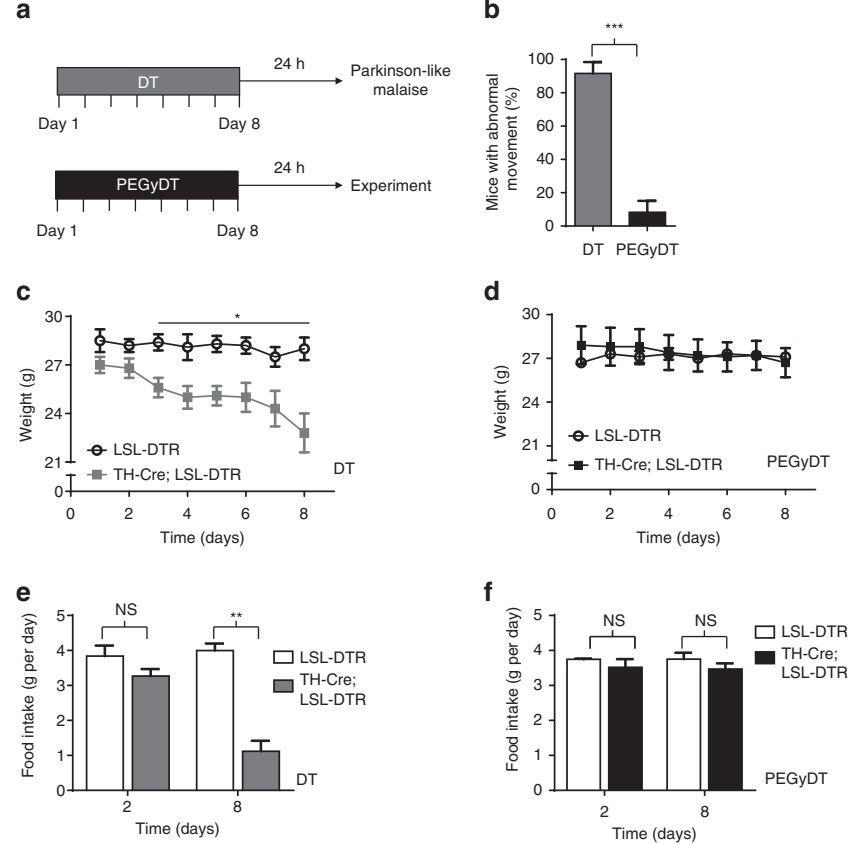

**Figure 3 | BRAINSPAReDT prevents Parkinson-like movements and decaying health.** (**a**) Schematic representation of the experimental design. (**b**) Scoring for abnormal movement of DT and PEGyDT-injected mice (***$P < 0.0001$, $n = 4$). (**c**) Mice weight during daily administration of 0.02 pmol g$^{-1}$ of body weight of DT (*$P < 0.01$, $n = 4$). (**d**) Mice weight during daily administration of 0.02 pmol g$^{-1}$ of body weight of PEGyDT ($n = 5$). (**e**) Food intake on the first and last day of administration of 0.02 pmol g$^{-1}$ of body weight of DT (**$P < 0.001$, $n = 3$). (**f**) Food intake on the first and last day of administration of 0.02 pmol g$^{-1}$ of body weight of PEGyDT ($n = 3$). Statistics were performed using unpaired $t$-test. Data are represented as mean ± s.e.m. NS, not significant.

**Regional sympathectomy predisposes obesity.** For more complete metabolic phenotyping, regionally sympathectomized mice were challenged with high-fat diet (HFD). On a HFD, mice that were regionally sympathectomized with BRAINSPAReDT gain more weight compared to controls (Fig. 5a,b, **$P < 0.001$, ***$P < 0.0001$, unpaired $t$-test) compared to controls: *TH-Cre; LSL-DTR* mice weight was 39.43 ± 1.83 g after 4 weeks on HFD, while control *LSL-DTR* mice weight was only 25.23 ± 1.42 g after the same time (the schematic timeline of injections and HFD is shown in Supplementary Fig. 5a)[13]. The increase in BW was due to a significant increase in fat mass (63% increase), which was accompanied by a decrease in lean mass (13% decrease, Fig. 5c). The speed of weight gain was also significantly higher for *TH-Cre; LSL-DTR* mice than for control *LSL-DTR* mice (**$P < 0.001$, unpaired $t$-test). While control mice gained 9.23 ± 0.72 g per month, *TH-Cre; LSL-DTR* mice gained nearly threefold (22.18 ± 1.06 g per month) in BW (Supplementary Fig. 5b). As the BRAINSPAReDT ablation is temporally controlled to take place in adulthood, no changes in body length were observed (Supplementary Fig. 5c).

Notably, mice sympathectomized with BRAINSPAReDT remained obese after withdrawal of HFD (Fig. 5d, ***$P < 0.0001$, unpaired $t$-test). After 40 days of HFD withdrawal, regionally sympathectomized mice weighed 38.20 ± 2.87 g, which was 35.46% higher than the initial weight before HFD. However, control mice reverted back to baseline, weighting only 2% more than the initial weight before HFD (Fig. 5d). We questioned

whether the sustained elevation in BW was caused by alterations in food intake. We verified that this is not the case, as food intake is ∼3 g per day for both experimental and control mice (Fig. 5e). Thus, the ratio of BW and food intake translated into higher feed efficiency in regionally sympathectomized mice, relative to control mice (Supplementary Fig. 5d).

Moreover, mice sympathectomized with BRAINSPAReDT displayed glucose intolerance, when on a normal diet (ND) regimen, contrasting with the normal glucose tolerance test of *LSL-DTR* control mice, which were also injected with PEGyDT (Fig. 5f, **$P < 0.001$, unpaired $t$-test).

The previous sympathectomy drugs, such as 6-hydroxydopamine neurotoxin, were known to increase catecholamines production, which normalizes after 2 weeks[14]. Thus, we also controlled for this temporal window, and challenged PEGyDT-ablated mice with HFD only 2 weeks after regional SNS ablation, to allow for normalization of hypothetical elevation of catecholamine levels. We verify that differences in BW were the same when mice were fed with HFD right after the ablation procedure or only 2 weeks after regional SNS ablation (Supplementary Fig. 5e **$P < 0.001$, unpaired $t$-test).

**Obesity is coupled to low thermogenesis and hypoactivity.** Similar to what was done for the cold challenge, we measured NE content and thermogenic gene expression in the adipose tissue of mice fed an obesogenic diet. Mice that were sympathectomized

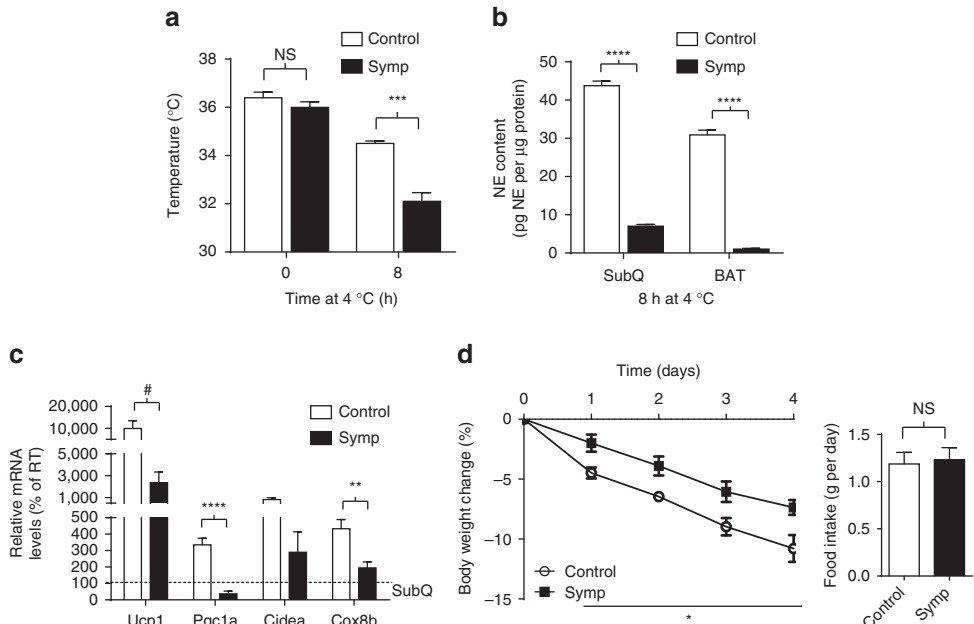

**Figure 4 | Sympathectomy with BRAINSPAReDT impairs adaptive thermogenesis and NE content in adipose tissues.** (**a**) Body temperature during cold (4 °C) exposure (\*\*\*P < 0.0001, n = 5). Controls are PEGyDT-injected *LSL-DTR* mice, which are hereafter referred to as 'Control'. (**b**) Norepinephrine content of SubQ adipose tissue and BAT after 8 h of cold exposure (\*\*\*\*P < 0.00001, n = 4 for Symp and n = 6 for Control). (**c**) mRNA levels of browning genes in SubQ adipose tissue after 8 h of cold exposure (#P = 0.066, \*\*P < 0.001, \*\*\*\*P < 0.00001, n = 9). (**d**) Body weight change and food intake during a leptin challenge (\*P < 0.01, n = 4). Statistics were performed using unpaired *t*-test. Data are represented as mean ± s.e.m. (related to Supplementary Fig. 4b,c). NS, not significant.

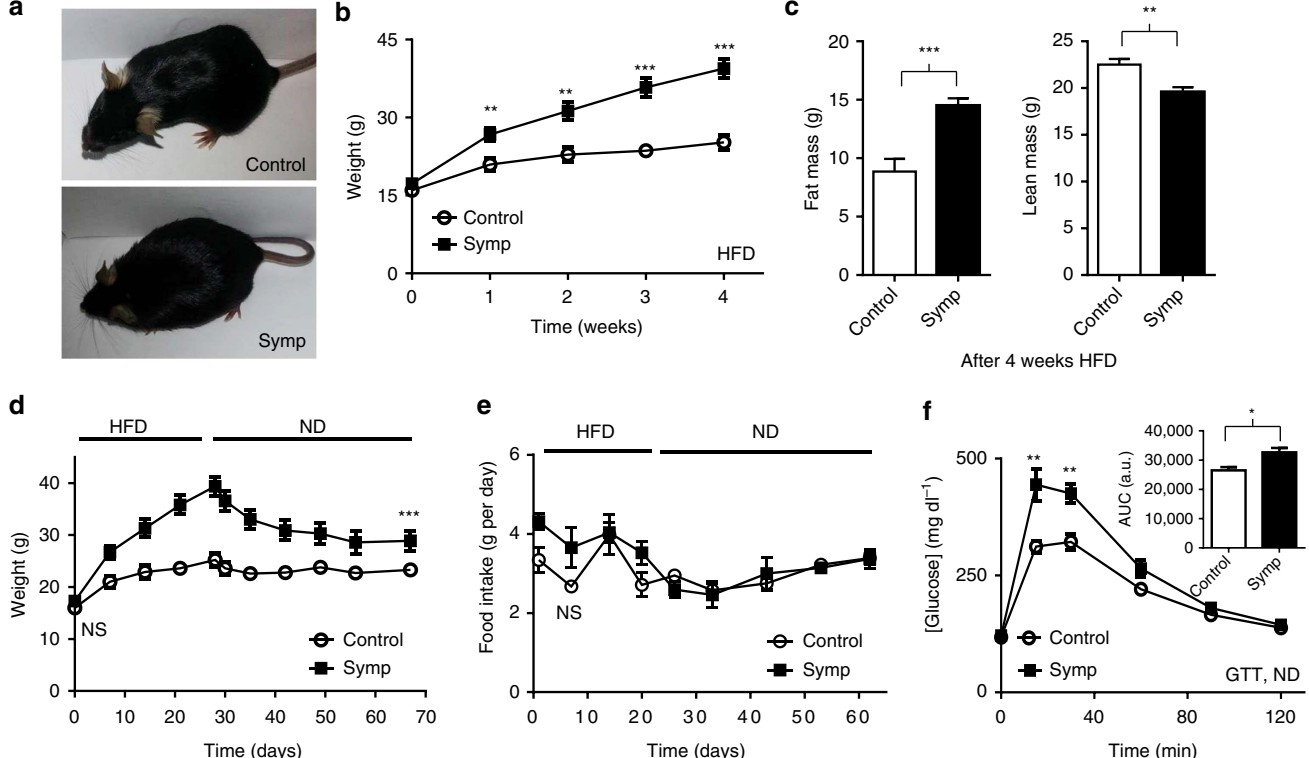

**Figure 5 | Sympathectomy with BRAINSPAReDT predisposes mice to sustained obesity and glucose intolerance without affecting food intake.** (**a**) *TH-Cre; LSL-DTR* and control *LSL-DTR* mice after 4 weeks of HFD regimen. (**b**) Weight gain during 4 weeks on HFD, after ablation (\*\*P < 0.001, \*\*\*P < 0.0001, n = 6). (**c**) Nuclear magnetic resonance analysis of fat and lean mass (\*\*\*P < 0.0001, \*\*P < 0.001, n = 8). (**d**) Weight variation before and after HFD withdrawal (\*\*\*P < 0.0001, n = 6). (**e**) Food intake before and after HFD withdrawal. (**f**) GTT during ND (\*P < 0.01, \*\*P < 0.001, n = 8). Statistics were performed using unpaired *t*-test. Data are represented as mean ± s.e.m. (related to Supplementary Fig. 5). GTT, glucose tolerance test.

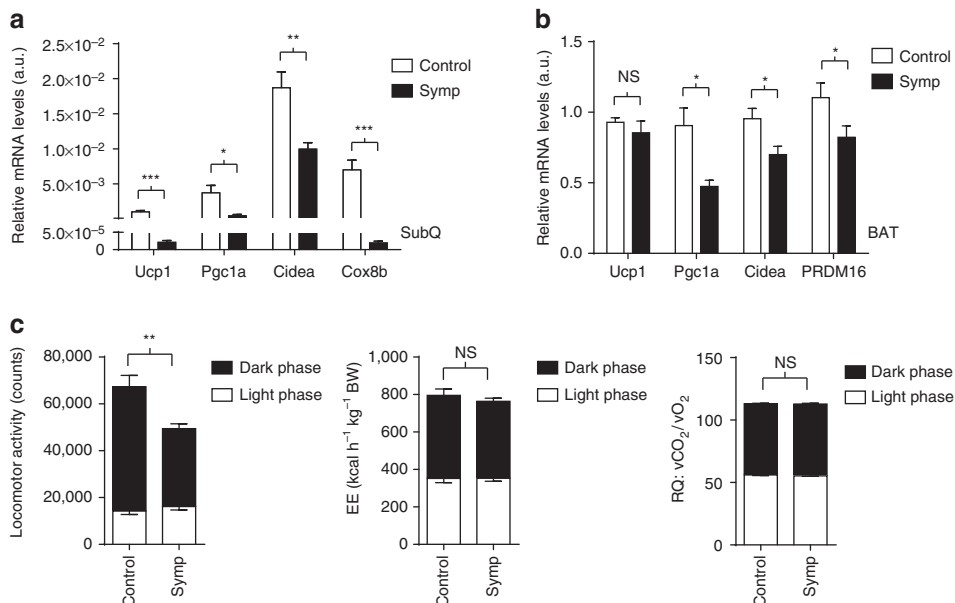

**Figure 6 | Obese mice sympathectomized with BRAINSPAReDT have deficits in thermogenic gene expression and nocturnal hypoactivity but normal energy expenditure.** (**a**) Thermogenic gene expression in SubQ adipose tissue (*$P < 0.01$, **$P < 0.001$, ***$P < 0.0001$, $n = 6$). (**b**) Thermogenic gene expression in BAT (*$P < 0.01$, $n = 7$ for Symp and $n = -8$ for Control). The a.u. used in **a,b** is the same. The data were normalized to GAPDH and SYBR Green was used as a probe. (**c**) Locomotor activity, energy expenditure and respiratory quotient (**$P < 0.001$, $n = 8$). Statistics were performed using unpaired $t$-test. Data are represented as mean ± s.e.m. (related to Supplementary Figs 6 and 7). EE, energy expenditure; NS, not significant; RQ, respiratory quotient.

with BRAINSPAReDT had an NE content (pg µg$^{-1}$ of protein) in SubQ WAT that was significantly lower than that of controls (38.71% decrease, ****$P < 0.00001$, unpaired $t$-test, Supplementary Fig. 4b). Moreover, and consistent with a lower NE tone in fat, we observed lower levels of thermogenic gene expression not only in this tissue but also in BAT (Fig. 6a,b; Supplementary Fig. 6a). In addition, UCP1 levels are also decreased in BAT of mice sympathectomized with BRAIN-SPAReDT (Supplementary Fig. 6b). Notably, total protein content was increased in BAT (as well as in SubQ WAT and visceral WAT; Supplementary Fig. 7a–c), indicating that the total UCP1 content in the tissue, which is the most relevant parameter for non-shivering thermogenic capacity[15,16], is even more decreased in regionally sympathectomized mice (UCP1/protein BAT (a.u.) is 6.0 ± 0.36 for controls and 4.9 ± 0.24 for regionally sympathectomized; *$P < 0.01$ and **$P < 0.001$, unpaired $t$-test). Finally, and for more complete metabolic phenotyping, we kept obese mice and controls in metabolic cages for 3 days (after 4 weeks on HFD). On HFD, mice that were sympathectomized with BRAINSPAReDT had lower levels of nocturnal locomotion compared to controls (Fig. 6c; Supplementary Fig. 7d–f). However, the energy expenditure and the respiratory quotient remained normal in mice that were sympathectomized with BRAINSPAReDT (Fig. 6c; Supplementary Fig. 7d–f).

## Discussion

The conditional Cre; LSL-DTR system is widely used for tissue-specific loss of function and genetic ablation of cell lineages[6]. The use of this system is contingent on a tightly regulated tissue-specific expression of Cre driver lines. Brain expression of peripheral Cre drivers may reflect not only insertion artefacts that dysregulate the transgene, but may also reflect cell lineage or biological function when the same gene is expressed in peripheral and central populations of cells.

Brain expression in many peripheral Cre drivers is a recurrent problem that has been limiting the use of the Cre; LSL-DTR system for ablating peripheral tissues, due to off-target effects. For instance, TH is an enzyme in the biosynthetic pathway of catecholamines, which include dopamine and NE. TH has been used as a marker and molecular handle to genetically manipulate catecholaminergic neurons with optogenetic techniques, which have a localized action[5,17–19]. The *TH-Cre* driver line labels dopaminergic neurons in the brain, as well as sympathetic neurons. As such, whole-body applications that require the systemic delivery of a toxin, like in the Cre; LSL-DTR system, have limited power because DT crosses the BBB.

To address this general problem, we developed a chemical genetic approach, named BRAINSPAReDT, in which we chemically modified DT by PEGylation (PEGyDT), so that it spares an action in DTR-expressing cells in the brain.

PEG is a non-toxic, non-immunogenic and non-antigenic biocompatible polymer that has been used as an excipient in many pharmaceutical and cosmetic formulations[8,9,20]. PEGylation increases hydrodynamic radius, polarity and molecular weight[21].

Permeability across the BBB is contingent on polarity and molecular weight, favouring non-polar and smaller molecular weight molecules[22]. Increased molecular weight and polarity is a known problem for CNS drug delivery, but a unique advantage for keeping DT away from the brain[23]. Thus, as shown here, increasing the polarity and the molecular weight of DT by PEGylation is a feasible way to prevent DT's action in the brain.

PEGyDT had an average increase in molecular weight by 436 Da, but it was just as efficient as DT in mediating tissue-specific cell death, both *in vitro* and *in vivo*. The latter was herein validated by analysing a Cre driver line, labelling populations of neuronal cells in the brain and in the periphery. We verified that genetic ablation of catecholaminergic neurons in the CNS led to a Parkinson-like phenotype, which is a major impairment for analysing the effect of regional SNS loss of function in a metabolic/obesity context. Parkinson-like mice have deteriorating health that leads to an emaciated state, which could be confused

with healthy weight loss. Alternatively, PEGyDT-mediated genetic regional ablation of the SNS spared catecholaminergic neurons in the CNS, and created a suitable rodent model system with which to study SNS function.

Our previous studies only probed the local role of SNS fat innervation in mediating lipolysis locally, in a single fat pad[5]. Here we probed whether this effect generalizes to systemic levels such as to control whole-body adiposity and metabolism. To achieve this, we performed a regional genetic loss of function of the SNS, that is, a regional genetic sympathectomy. Sympathectomy procedures that have been used in the past included chemical ablation with guanethidine or 6-hydroxydopamine[24–27]. Despite the experimental differences, we are able to draw parallels between classical studies and our findings, as follows: if not challenged by a HFD or cold, mice treated with BRAINSPAReDT present normal BW, and other metabolic parameters, which is consistent with results reported in classical studies involving chemical sympathectomy[24,25]. When challenged with a HFD, we then see an effect in obesity and thermogenesis that is similar to results previously reported[27]. However, it was also reported that mice chemically sympathectomized with guanethidine do not become obese after a human supermarket diet[26]. Of note, another study disclosed that guanethidine treatment lead to pruritus, open sores and skin necrosis[25]. We also observed that chemical sympathectomy caused malaise (data not shown). Finally, chemical ablation with 6-hydroxydopamine is classically used as a parkinsonian agent, and thus ablates beyond the SNS, affecting dopaminergic neurons in the brain[28]. BRAINSPAReDT does not cause any malaise or side effects, due to its tissue specificity and deficient access to the brain.

Studies using surgical techniques (ganglia removal or axotomy) reported findings similar to ours in several species (rats, rabbits, cats and hamsters). Several of these studies report localized fat accumulation and impaired fat mobilization[2–4].

Taking the classical literature and our results into account, genetic ablation via BRAINSPAReDT, with expression of Cre-inducible DTR and PEGyDT injection, is an amenable way to combine tissue specificity for the SNS, and a regional action enabled by the Cre driver line. We used a sympathetic Cre driver that has a mosaic pattern of expression, which preferentially marks SNS neurons innervating fat and avoids other organs such as heart and lungs. The mosaicism of this *TH-Cre* driver mouse line may be due to insertional effects of the transgene, which alter the activity of the promoter in a tissue-specific manner. This restricted Cre expression within the SNS enabled regional sympathectomy with BRAINSPAReDT, rendering mice extremely vulnerable to metabolic challenge, while compromising several key elements in mechanism for energy dissipation. Consistent with the known role of NE in driving thermogenesis, mice sympathectomized with BRAINSPAReDT have deficient adaptive thermogenesis, which is accompanied by a down-regulation of thermogenic genes and NE content in adipose tissues. In addition, sympathectomy with BRAINSPAReDT rendered mice extremely prone to metabolic syndrome and obesity, as seen during a HFD challenge.

Mice regionally sympathectomized were glucose intolerant and did not normalize excess weight upon HFD withdrawal, as it would be expected. Instead, increased weight plateaued upon HFD withdrawal. Whereas we cannot exclude a role of the adrenals in the phenotype we see, leptin action does not depend on intact adrenal glands[29]. Thus, the persistence of adiposity in regionally sympathectomized mice, as well as the defective responses to leptin, is consistent with the known role of sympathetic neuro-adipose junctions in driving lipolysis and fat mass reduction[5]. The lack of WAT mobilization in regionally

sympathectomized mice is consistent with the lack of adaptive thermogenesis. Finally, under HFD, regionally sympathectomized mice have reduced nocturnal activity, which further reduces energy dissipation. As a consequence of regional SNS loss of function, mice appeared to have sustained obesity, even though eating behaviour was normal.

Future studies will attempt to develop versions of DT that can ablate the innervation to specific organs or specific peripheral tissues/cell types. This may be achieved by using Cre driver mice, where the expression is restricted to neurons innervating specific organs.

While consolidating the appetite-independent link between adiposity and the SNS, our results provide a proof of principle that BRAINSPAReDT can generally be used for Cre/DTR tissue-specific ablation outside the brain using CNS Cre drivers. For instance, several subpopulations of immune cells express neurotransmitter or CNS receptors. However, the functional role of such neuro-like immune cells remains elusive[30,31]. In this regard, BRAINSPAReDT can now be used for loss-of-function studies using CNS Cre drivers. Overall, BRAINSPAReDT will promote the advancement of emerging fields of research such as neuroimmunology, which lay at the interface between neurobiology and physiology.

## Methods

**PEGylation reaction.** An amount of 1 mg of DT *Corynebacterium diphtheriae* (Calbiochem) was reconstituted in 1 ml of conjugation PBS 1x (concentration 17 µM). Modification was performed using 80 equiv. of modifying agent in relation to DT. Hence, 0.001269 mmol of crosslinker was needed. Mass spectrometry (PEG)₄ methyl-PEG-NHS-Ester (Life Technologies), after being fully equilibrated to RT, was reconstituted in 1.1 ml of dimethyl sulfoxide (Sigma-Aldrich), leading to a concentration of 250 mM PEG, according to the manufacturer's instructions. A concentration of 250 mM PEG was diluted 1:100 in dimethyl sulfoxide and then the appropriate volume was transferred into DT solution. The mixture was incubated for 4 h, at RT, under shaking. To remove the excess of crosslinker, gel filtration chromatography using the gravity protocol of PD MidiTrapG-25 column (GE Healthcare) was performed according to the manufacturer's indications. Briefly, the column was prepared and equilibrated with buffer. The sample was applied to the column, the flow-through was discarded, and the eluate was collected. The concentration of PEGyDT was determined using a NanoDrop ND-2000 UV-Visible spectrophotometer (Fisher Scientific), measuring the absorbance at 280 nm of 1 µl samples and using PBS buffer as blank.

**Mass spectrometry.** DT and PEGyDT samples were analysed by MALDI–TOF. Samples were previously concentrated in Amicon Ultra-2, Ultracel-10 membranes (Sigma-Aldrich) and prepared according to the manufacturer's instructions (centrifuged for 4 min, 3,500*g*, with PBS 1x). DT and PEGyDT were centrifuged at 3,500*g* until appropriate concentration (10 µM).

Samples were desalted using a Millipore C4 Ziptip and eluted with 2 µl of 50% MeCN/0.1%TFA. A measure of 0.8 µl of this sample was mixed with 0.8 µl 2′,5′-dihydroxyacetophenone matrix (Fluka, 15 mg ml⁻¹ in 75% EtOH containing 0.075 M ammonium citrate) and 0.8 µl 2% trifluoroacetic acid. The sample was air-dried onto a target plate and analysed in a Water MALDI microMX mass spectrometer at threshold laser power in linear positive ionization mode. Pulse–voltage settings were optimized for maximum resolution of the 2 + charge-state peaks. External calibration was with a mixture of equine myoglobin and bovine trypsinogen. Data were analysed using Waters Masslynx4.1 software.

**Cell culture of HeLa cells.** HeLa cells were obtained from the laboratory of Lars Jansen at Instituto Gulbenkian de Ciência (IGC) and maintained in Dulbecco's modified Eagle medium high glucose with L-glutamine and sodium pyruvate (Biowest) supplemented with 10% fetal heat-inactivated (Biowest) and 1% penicillin–streptomycin (Biowest)—Dulbecco's modified Eagle medium complete medium. Cells were maintained in humidified atmosphere, 5% $CO_2$ at 37 °C. Cells were counted in a haemocytometer (Neubauer Cell with Double Grid, Fisher Scientific) using 1% trypan blue solution (Sigma-Aldrich) and plated at 10⁵ and 6 × 10⁴ cells per well in 12- and 24-well plates (CORNING), respectively.

All incubations used a concentration of 6.50 nM of DT and PEGyDT. Different time points (6–72 h) were prepared for FACS analysis. The medium and cells were collected and cells were centrifuged for 4 min, 3,250*g*, at 4 °C. Cells were resuspended in 150 µl FACS buffer (2% FBS, 0.02% sodium azide-1-¹⁵*N* (Sigma-Aldrich) in PBS 1x) and 25 µl propidium iodide (Sigma-Aldrich) were added.

Data were acquired in a Becton Dickinson Flow Cytometer FACSCalibur. Data were acquired using CellQuest software (Becton Dickinson) and analysed in FlowJo.

**Mice and housing conditions.** Mice (male and female) 12–16 weeks old were housed at controlled temperature and humidity, under a 12 h light/dark cycle. Food and water were supplied *ad libitum*, unless mentioned otherwise. The animal experiments were performed in agreement with the International Law on Animal Experimentation and were approved by the IGC ethics committee and by the USC Ethical Committee (Project ID 15010/14/006). All controls were injected with vehicle (PBS 1x). C57BL/6 mice were obtained from the Mice Production Facility at IGC. *TH-Cre* (#008601), *LSL-DTR* (#007900), *LSL-ChR2-YFP* (#012569) and *GFPL10* (#024750) mice were purchased from Jackson Laboratory.

**PEGyDT-mediated regional sympathectomy.** *TH-Cre; LSL-DTR* mice were used for this experiment and *LSL-DTR* mice were used as controls. During the experiment, animals and food were weighed every day. Both DT and PEGyDT were injected i.p. Injections of 0.02 pmol g$^{-1}$ of BW of DT or PEGyDT were administered once a day for 8 consecutive days.

Mice were perfused with PBS 1x followed by 4% formaldehyde (VWR). The brain was removed and fixed in 4% formaldehyde for 2 days, at 4 °C. Using a vibratome VT 1000S (Leica), 50 μm-thick slices of ventral tegmental area, area postrema, locus coeruleus, hypothalamus and substantia nigra were obtained. These slices were at RT for 1 h, shaking in blocking buffer (1% bovine serum albumin, 0.1% Tween 20 (Sigma-Aldrich), 0.05% sodium azide in PBS 1x). The slices were incubated with the primary antibody TH chicken (1:1,000, Aves Lab, #TYH) overnight shaking, at 4 °C. After washing, secondary antibody Alexa Fluor 488 Goat Anti-chicken IgY (H + L; 1:500, Life Technologies, #A-11039) was applied for 1 h, shaking, at RT. After secondary antibody incubation, slices were washed and mounted with Dapi-Fluoromount-G (Southern Biotech). The slides were dried at RT and then kept at 4 °C.

Fibres removed from inguinal fat were fixed in 2% formaldehyde for 2 h, at RT, shaking. After blocking for 1 h, at RT, fibres were incubated with primary antibodies rabbit polyclonal to neuron specific β$_3$-Tub (1:1,000, Abcam, #ab18207) and TH chicken, overnight, shaking, at 4 °C. After washing, secondary antibodies Alexa Fluor 488 Goat Anti-rabbit IgG (H + L; 1:500, Life Technologies, #A-11008) and Alexa Fluor 594 Goat Anti-chicken were applied for 1 h, shaking, at RT. After secondary antibodies incubation, fibres were washed and mounted with Fluoromount-G. The slides were dried at RT and then kept at 4 °C.

Confocal images were acquired with Leica TCS SP5 Inverted and Upright Microscopes. Analysis and quantification of acquired images were performed using FIJI.

**Scoring test.** After administration of 0.02 pmol g$^{-1}$ of BW of DT or PEGyDT once a day for 8 consecutive days, movies of mice were done. These movies were presented to people blinded to the experiment, but experienced in working with mouse models. The question 'How many animals are moving normally?' was asked.

**Functional tests on regionally sympathectomized mice.** Body rectal temperature was measured using an electronic thermometer (Precision) when the animals were housed at RT and animals were weighed. After this, animals were housed at 4 °C with ND food and water *ad libitum*. Body temperature was measured and animals were weighed every hour for 8 h.

Mice were given 4 daily i.p. injections of either 2 mg kg$^{-1}$ leptin (Amylin Pharmaceuticals Inc.) or PBS. BW and food intake were measured daily.

Animals were weighed and ND was replaced by HFD. Mice were weighed every week for 4 weeks. HFD (Ssniff, Spezialdiäten GmbH, Soest, Germany) contains 60 kJ% fat. After 22 days, HFD was replaced by a ND regimen and, during this period, mice were weighed and food intake was measured.

After 12 h fasting during the dark cycle, glucose was measured using Accu-Chek Aviva Glucose (Roche Diagnostics). After this, an i.p. injection of 2 mg g$^{-1}$ D-(+)-glucose (Sigma-Aldrich) was administered and glucose level was measured at 15, 30, 60, 90 and 120 min post injection. These procedures were performed on ND. Nose-to-anal distance was measured.

ND was replaced by HFD 2 weeks after regional sympathectomy was performed in *TH-Cre; LSL-DTR* mice and also in control *LSL-DTR* mice. Food intake and weight were measured.

**NE measurements (enzyme-linked immunosorbent assay).** To assess the basal content of NE in innervated organs, 12-week-old *LSL-DTR* and *TH-Cre; LSL-DTR* mice were killed for tissue collection 1 week after 8 doses of PEGyDT (once a day). To examine the effect of cold exposure (8 h) and of HFD (4 weeks) on the levels of NE in WAT after regional sympathectomy, subcutaneous fat pads were collected for homogenization right after the challenges. NE levels were determined with an NE ELISA kit (Labor Diagnostika Nord GmbH). Tissues were homogenized and sonicated in homogenization buffer (1 N HCl, 1 mM EDTA), and cellular debris were pelleted by centrifugation at 16,000g for 15 min at 4 °C. All tissue samples were normalized to total tissue protein concentration.

**Quantitative PCR.** Total RNA was isolated from either superior cervical ganglia or SubQ fat pads using RNeasy Mini kit (Qiagen), from which complementary DNA was reverse-transcribed using SuperScript II (Invitrogen) and random primers (Invitrogen). Quantitative PCR was performed using SYBR Green (Applied Biosystems) in ABI QuantStudio 7 (Applied Biosystems). Glyceraldehyde 3-phosphate dehydrogenase (GAPDH) housekeeping gene was used to normalize samples. The list of primers used is shown in Supplementary Table 1.

Total RNA in BAT was isolated by using Trizol Reagent (Invitrogen) according to the manufacturer's protocol (RNA was precipitated with chloroform and isopropanol, washed with 75% ethanol, and finally dissolved in RNAse-free water). Complementary DNA synthesis was performed with MMLV enzyme (Invitrogen) following supplier protocol. Real-time PCR (TaqMan; Applied Biosystems) was performed using specific sets of primers, shown in Supplementary Table 1. All reactions were carried out using the following cycling parameters: 50 °C for 2 min, 95 °C for 10 min followed by 40 cycles of 95 °C for 15 s, 60 °C for 1 min[32]. For PRDM16 assay, ID Mm01266512_m1 (Applied Biosystems TaqMan Gene Expression) was used. Values were expressed relative to hypoxanthine–guanine phosphoribosyltransferase levels.

**Nuclear magnetic resonance analysis.** For the measurement of body composition, nuclear magnetic resonance imaging (Whole Body Composition Analyzer; EchoMRI; Houston, TX) was used. EchoMRI is a body composition analyser for live subjects that measures body fat and lean masses within short scan times (to keep the measurement as comfortable as possible for the animals). Mice were not under anaesthesia and no special preparation was needed before the measurements. Mice were placed in a holder of custom-defined size during the measurement (measuring time ranged between 0.5 and 3.2 min)[32–34].

**Calorimetric measurements.** Mice were analysed for locomotor activity, energy expenditure and respiratory quotient using a calorimetric system (LabMaster; TSE Systems; Bad Homburg). This system is an open-circuit instrument that determines O$_2$ consumption, CO$_2$ production and respiratory quotient (vCO$_2$/vO$_2$)[32–34]. Mice were acclimated for 1 week before starting the measurements.

**Thoracic sympathetic ganglia analysis.** Thoracic sympathetic ganglia (T1–12) were dissected from *TH-Cre; LSL-ChR2-YFP*[35] and T13 was dissected from *TH-Cre; LSL-GFPL10* (ref. 36) 6–8-week-old mice, fixed in 4% formaldehyde for 2 h and stained using anti-TH (1:1000, Pel-freez, #P40101-150) and anti-GFP (1:1000, Abcam, #ab13970) antibodies. Z-series stacks were acquired on a Leica TCS SP5 confocal Inverted microscope. Analysis and quantification of images were performed in FIJI.

**Western blotting.** Dissected BAT was homogenized and lysed with buffer with the following composition: Tris-HCl pH 7.5 50 mM, EGTA 1 mM, EDTA 1 mM, Triton X-100 1% v/v, sodium orthovanadate 0.1 mM, sodium fluoride 50 mM, sodium pyrophosphate 5 mM, sucrose 0.27 M and protease inhibitor cocktail (Roche Diagnostics)[37]. The protein concentration was determined by the Bradford method. Protein lysates were subjected to SDS–polyacrylamide gel electrophoresis, electrotransferred on a polyvinylidene difluoride membrane and probed with the following antibodies: UCP1 (1:10,000, Abcam, #ab10983) and α-tubulin (1:5,000, Sigma, T5168)[32–34]. Then, the membrane was incubated with the corresponding secondary antibody: anti-mouse or anti-rabbit (all of them from DAKO, Denmark A/S). The membranes were exposed to an X-ray film and developed using Developer (Developer G150) and Fixator (Manual Fixing G354). The quantification was done with the ImageJ program. Values were expressed in relation to α-tubulin protein levels.

**Protein quantification.** BAT, SubQ and visceral dissections were weighed and homogenized in lysis buffer (consisting of a mix of 5 ml Tris-HCL, 0.5 ml 0.2 M EGTA, 0.5 ml 0.2 M EDTA, 1 ml Triton X-100, 1 ml 0.1 M sodium orthovanadate, 0.21 g sodium fluoride, 0.22 g sodium pyrophosphate and 9.2 g of 0.27 M sucrose, made up with distilled water to 100 ml and adjusted to pH 7.5) and freshly added protease inhibitor cocktail tablets (Roche Diagnostics). The protein concentration was determined by the Bradford Method, and the total protein content of the tissues was calculated[32–34].

**Statistical analysis.** All the statistical analysis were performed in Sigma Plot 11.0, using unpaired *t*-test when two groups were being compared or one-way ANOVA test (followed by Tukey test) when several groups were being compared (more specifically in Figs 1 and 2). A *P*-value < 0.05 was considered as statistically significant. Data were represented as mean ± s.e.m.

**Data availability.** The data that support the findings herein presented are available from the corresponding author upon reasonable request.

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

## Acknowledgements

This work was supported by the Fundação para a Ciência e Tecnologia (PTDC/BIM-MET/3750/2014), the European Molecular Biology Organization (EMBO), European Community's Seventh Framework Programme (FP7/2007-2013) under agreement no. 281854—the ObERStress project, MINECO co-funded by the FEDER Program of EU (ML: SAF20015-71026-R) and Xunta de Galicia (ML: 2015-CP079). The FCT, IGC and EMBO supported M.M.A.P., I.M., E.S., N.K., R.M.P. and A.I.D. G.J.L.B. is a Royal Society University Research Fellow and a FCT Investigator. We thank Dr Len Packman for the mass spectrometry services and Professor Carlos Romão for useful discussions. We thank Colin Adrain and Lars Jansen for critical reading of the manuscript.

## Author contributions

A.I.D. conceived the experimental strategy; G.J.L.B. conceived the protein modification; P.C. provided intellectual input on thermogenesis; M.O.D. contributed to the calorimetric studies. PEGylation was performed by M.M.A.P.; cell culture and FACS were conducted by E.S. and M.M.A.P.; M.M.A.P. developed PEGyDT-mediated sympathectomy; M.M.A.P., E.S., N.K. and R.M.P. performed functional tests; I.M. conducted quantitative PCR and NE measurements; N.M. and M.L. performed nuclear magnetic resonance, locomotor activity, energy expenditure and respiratory quotient analysis; N.M. and M.L. performed PCR with reverse transcription and western blotting in BAT as well as protein quantification; rodent husbandry was performed by N.K.; A.I.D. and M.M.A.P. wrote the manuscript; all authors revised the manuscript. G.J.L.B. and A.I.D. are co-senior authors of this work.

## Additional information

**Competing interests:** The authors declare no competing financial interests.

DOI: 10.1038/ncomms15673    **OPEN**

# Corrigendum: A brain-sparing diphtheria toxin for chemical genetic ablation of peripheral cell lineages

Mafalda M.A. Pereira, Inês Mahú, Elsa Seixas, Noelia Martinéz-Sánchez, Nadiya Kubasova, Roksana M. Pirzgalska, Paul Cohen, Marcelo O. Dietrich, Miguel López, Gonçalo J.L. Bernardes & Ana I. Domingos

*Nature Communications* 8:14967 doi: 10.1038/ncomms14967 (2017); Published 3 Apr 2017; Updated 17 May 2017

The financial support for this Article was not fully acknowledged. The Acknowledgements should have included the following:

[***Human Frontiers Science Program (HFSP) funds the labs of A.I.D. and P.C. ***].

