## [Peer Review File · Nature Communications]

1st round of revisions

Reviewers' comments:

Reviewer #1 (Expert in DA neuron function; Remarks to the Author): This manuscript addresses an interesting technical problem with the application of DT transgenic technology for ablation of peripheral neurons selectively.

A: We appreciate the reviewer's enthusiasm for our new method.

Reviewer #1 Standard PEGylation of DT does not affect toxicity, but limits BBB penetration.

A: That is an accurate account of our method.

Reviewer #1 Overall the technical aspect of these findings would seem useful but of limited interest to a broad audience.

A: Our genetic method can be used to ablate any population of genetically identified peripheral cells, that can be labeled with a CNS Cre driver line. This methodological approach generalizes to broad research areas of Neuroscience, Metabolism or Immunology. In this regard, we have tested BRAINSPAReDT with a myeloid Cre driver mouse line, which labels macrophages and microglia in the brain. Intraperitoneal injection of BRAINSPAReDT enabled the first ablation of macrophages that spares microglia in the brain (unpublished data not shown). We hope that these results will mitigate the Reviewer's concern, for perceiving our study to have "*limited interest to a broad audience*".

Reviewer #1 This approach is then used to eliminate peripheral sympathetic innervation and the mice display robust defects in maintaining body temperature and weight maintenance, as well as reduced activity. However, these observations are not really pursued mechanistically any further.

A: We agree with the reviewer's assessment and provide additional data in Figure 4d that supports a mechanism by which the SNS mediates the weight reducing effects of leptin. A leptin challenge was performed (for 4 days) in order to address this question. As mentioned, the results are shown in Figure 4d.

Reviewer #1 Minor issue:

The following sentence is unclear: "We cannot exclude that effects on body temperature and weight are not due to ablation of SNS to tissues besides fat, like the adrenal gland, which also releases NE".

A: We appreciate this comment, and agree that sentence was unclear. Therefore, that sentence has been deleted from the new version of the manuscript

Reviewer #2 (Expert in sympathectomy; Remarks to the Author): Your paper reports development of a novel method for producing sympathectomy without adversely affecting catecholaminergic neurons in the brain, based on the use

of a non brain penetrating conjugated diphtheria toxin (BRAINSPAReDT) and conditional expression of diphtheria toxin receptor in sympathetic neurons. The approach is a clever one and does appear to spare brain neurons as you had expected. The effects of this treatment on thermoregulation and fat accumulation in mice are quite profound and well-documented experimentally. The promise of refining the technique further to allow tissue specific sympathectomy is very exciting.

A: We sincerely appreciate the reviewer's enthusiasm for our paper and we agree that this novel technique awaits exciting new applications.

Reviewer #2 However, I have two very important concerns:

1) The BRAINSPAReDT procedure doesn't seem to cause sympathectomy when judged by the most common measure typically used to confirm sympathectomy, namely tissue NE content. Supplemental figure 3B shows that NE content was unaffected in many tissues (e.g. lung, liver, heart and gut) and only mildly reduced in others (spleen and kidney). The criterion used by most investigators is that "sympathectomy" requires at least 90% reduction in tissue NE content. By this standard, no tissues were actually sympathectomized in this study.

A: The reviewer brought to our attention a pertinent question that led us to perform additional experiments that are now found in Supplementary Figure 4c. We think that these new evidence explains the lack of effect on other organs such as heart and lungs. Given that the lung and the heart are innervated by the thoracic paravertebral SNS chain, we have characterized the TH-Cre driver mouse line by immunofluorescence in mice crossed with a reporter LSL-ChR2-YFP. We discovered incomplete co-localization of TH and ChR2-YFP reporter in the thoracic ganglia of the sympathetic chain. We see that only $59.62 \pm 4.22\%$ of TH⁺ neurons are YFP⁺. This explains the lack of decreased NE in the organs innervated by these ganglia. However, in SNS neurons in adipose tissue, the TH-Cre line proves to have complete co-localization of YFP and the marker TH (Figure 4A in Zeng et al. 2015). This complete overlap in fat is consistent with the marked decrease in NE content that we measure in fat (Figure 4b and Supplementary Figure 6a). Thus, we can conclude that, outside the brain, the TH-Cre driver line is a peripheral mosaic that marks neurons innervating fat, but not all SNS neurons innervating vital organs such as heart and lungs. The mosaicism of this TH-Cre driver mouse line may be due to insertional effects of the transgene, that alter the activity of the promoter in a tissue-specific manner. These results nicely surprised us, as we unexpectedly uncovered a molecular handle that preferentially labels the SNS innervation in fat. Hence, in the revised version of the manuscript, we claim that we have achieved regional sympathectomy. We are thus grateful to the reviewer for bringing this to our attention.

Reviewer #2 (Incidentally, I couldn't find the tissue NE content measurements in fat.)

A: NE was measured in fat and this was shown in the submitted version of the paper as well as in revised version, in Fig. 4b (Symp mice have $85.4 \pm 3.58\%$ less NE content compared to Control mice in the same condition) and in Supplementary Figure 6a, which was previously Supplementary Figure 5a (Symp mice have $90.2 \pm 0.86\%$ less NE content compared to Control mice in the same condition).

Reviewer #2 Although it's clear that some abnormalities of sympathetic function were produced by the BRAINSPAReDT procedure, that is a very different matter from sympathectomy.

A: We agree with the reviewer that we do not perform global sympathectomy, but rather, a regional sympathectomy. As explained above, we now introduce the term “regional sympathectomy”. In our case, we take advantage of the mosaicism of the TH-Cre driver line, which faithfully expresses in TH-expressing neuron in fat (Zeng et al. 2015), but not in thoracic organs. The intestine is also not affected by genetic ablation of SNS, which we conclude from the lack of significant difference in gut NE between sympathectomized and control mice. In the future, BRAINSPAReDT will enable organ-specific sympathectomies, by taking advantage of Cre-driver lines that mark the SNS innervation to specific organs. This “regional sympathectomy” could not possibly be achieved with chemical methods, and represents an advanced modality for performing loss of function of sub-compartments within the SNS.

Reviewer #2 2) The dramatic effect of BRAINSPAReDT on fat accumulation in overfed mice is very surprising in light of the fact that numerous previous workers have shown little or no effect of sympathectomy on obesity development in rats and mice (Am J Physiol. 1984 Dec;247(6 Pt 2):R979-87; Physiol Behav. 1985 Sep;35(3):455-63; Physiol Behav. 1985 Sep;35(3):473-7; Can J Physiol Pharmacol. 1991 Dec;69(12):1868-74). This previous literature needs to be thoroughly assessed and discussed. The obvious explanation is that BRAINSPAReDT is having important effects other than sympathectomy.

A: We appreciate that the reviewer brings this classical literature to our attention. We thoroughly analyzed these papers and compared them to our results, to conclude that some of these reports are consistent with our results. Despite the experimental differences in terms of species, age, food challenges, and unspecific side effects of some of the chemical agents (discussed below), we were still able to draw parallels between all of these classical studies and our findings. For instance, the reviewer mentions that our mice are “overfed”, but in our study mice treated with BRAINSPAReDT present normal food intake when compared to controls. This data is displayed in Figure 3f, Figure 5e and Supplementary Figure 4e of the previous version of the manuscript, which are the same in the revised version (except for Supplementary Figure 4e, that's is now Supplementary Figure 5e). Moreover, if not challenged by a HFD or cold, mice treated with BRAINSPAReDT present normal body weight, and other metabolic parameters. These results are consistent with those in two of the papers indicated by the reviewer: Am J Physiol. 1984 Dec;247(6 Pt 2):R979-87, and Physiol Behav. 1985 Sep;35(3):473-7. As performed in the latter paper, we also measured water intake and found an effect on water intake. SNS loss of function may decrease blood pressure, which is well known to activate the angiotensin-renin system and ADH in the hypothalamus, which in turn increases thirst sensation. In the

figure, it is shown the cumulative water intake during 24 h.

When mice are fed a HFD we then observed an effect on obesity, thermogenesis and UCP1 that is similar to the results reported in one of the papers that is referred by the reviewer: Can. J Physiol Pharmacol. 1991 Dec; 69(12):1868-74. The standard rodent HFD used in this latter paper, as well as in our study, may be nutritionally dissimilar to a human supermarket diet used in Physiol Behav. 1985 Sep;35(3):455-63. In this study, sympathectomized mice do not gain weight on a human supermarket diet, but these differences may be due to malaise or discomfort, which is reported page 474 of one of the papers that the reviewer cites (Physiol Behav. 1985 Sep;35(3):473-7): ***“Preliminary work revealed that guanethidine treatment produced skin necrosis and, subsequently, open sores as each treated rat attempted to scratch the injection site. Injections within IBAT eliminated this problem as the rats could not reach the injection site.”*** These mice could not reach the area to be scratched, but they may still feel the same pruritus. In the work of one of the authors of this manuscript, it was also experienced that these chemicals, when injected into adult mice, leads to severe malaise (data not shown). In this regard, it is very important to highlight that BRAINSPAReDT does not cause any malaise or side effects, due to its tissue specificity.

Reviewer #2 In addition to those major concerns, I have two suggestions for improvement of the paper.

1) You need to do a thorough literature review on previous methods used to produce sympathectomy and their relative strengths and weaknesses. The current discussion of this topic is both wrong (capsaicin is not used to produce sympathectomy) and hugely incomplete.

A: We appreciate the reviewer's intention for improving our manuscript. As suggested, we have now discussed the former literature in greater detail and draw parallels with the aforementioned classical papers.

Reviewer #2 2) The statistical analysis described is inadequate to the task of analyzing all of the different types of experimental designs in your study. Please provide more detail.

A: All the statistical analysis were performed in Sigma Plot 11.0, using unpaired t-test when two groups were being compared or One-Way ANOVA test (followed by Tukey test) when several groups were being compared (more specifically in Figure 1 and 2). A P value < 0.05 was considered as statistically significant. Data were represented as mean \pm standard error of the mean (SEM). These details were mentioned in the Materials and Methods Section.

Reviewer #3 (Expert in adipose tissue biology; Remarks to the Author): I have one point for clarification. In fig. 6 a and b, the data are given as a.u. - The authors should indicate in the figure legends whether the a.u. in a and b is the same, i.e. that the UCP1 mRNA levels in BAT are some 100 fold higher than in SubQ.

A: The figure legend was now clarified and we indicate that a.u. are the same in Figure 6a and in Figure 6b. The data was normalized to GAPDH and the same probe was used (SYBR Green).

In addition, we also provide UCP1 levels in BAT (Supplementary Figure 6b). The original western blots for UCP1 and α -tubulin are presented below:

Reviewer #3 Further, I lack some information on what effect the treatment has had on the total amount of BAT (and perhaps SubQ). Wet weight is not of much interest, but were the total protein amounts or the total RNA amounts obtained from the tissue dissections different in the control and the treated mice?

A: This is a pertinent question that is now addressed not only for BAT and SubQ, but also for visceral adipose tissue. The results are shown below and are included in Supplementary Figure 7a, b, c.

Reviewer #4 (Expert in protein PEGylation; Remarks to the Author): The modification of DT with PEG4-NHS esters appears to have been carried out correctly, as has the characterization of this variant via mass spectrometry. The authors should make it absolutely clear that PEGylated DT is a heterogeneous mixture of PEGylated isoforms that differ in the number, location and occupancy of PEGylation sites. The authors should also comment on the approximate fraction of DT that remains unmodified following the PEGylation reaction.

A: We thank the reviewer for these suggestions. We have added the following sentence into the manuscript for clarity:

"Conversion of DT to PEGyDT was nearly complete (relative 95% conversion) and resulted in a heterogeneous mixture differing in both the copy number of PEG units and conjugation sites."

In addition, we have added an additional Supplementary Figure (new Supplementary Figure 1) with the raw MALDI-TOF data. Also, we have annotated a peak on the mass spectrometry on Figure 1c of the product of the reaction between DT and NHS-PEG₄ showing that there is a small percentage of unmodified DT. We estimate the relative conversion to be 95%. However, we would like to note that MALDI-TOF is not a quantitative technique and this is why we have only an estimated conversion (this is based on the relative area of each peak).

2nd round of revisions

Reviewer's comments:

Reviewer #2 (Remarks to the Author):

Thank you for taking my previous critique seriously and responding with a significant re-write of the MS and a wholly new interpretation of your findings.

A: We appreciate the comments from the reviewer that have allowed us to improve our manuscript by providing additional details.

You now assert that the BRAINSPAReDT technique that you developed causes quite specific deletion of sympathetic neurons to adipose tissue, but not to other organs. This would be an exciting technical advance if true, but you have not provided convincing evidence.

First, you still do not report actual NE content in the various fat depots of the treated rats to confirm the extent of sympathectomy. The figures you cite only show percentages. It is critical that you report absolute values, preferably in Supplementary Figure 4b, where these values are reported in proper units for all other organs except adipose tissue.

A: We had expressed those values as percentages because the reviewer wanted to see whether BRAINSPAReDT leads to a 90% reduction of NE content in tissues (stated in comment #1 of previous review). We have however changed the units to comply with this new request. We now report in Supplementary Figure 4b the NE content of adipose tissue expressed in absolute values (pg NE per µg protein), as we did for all other organs. We do the same for other figures, such as Figure 4b, which plots NE content in SubQ and BAT after cold exposure.

Second, on page 9 you write:

"We discovered incomplete co-localization of TH and ChR2-YFP reporter in the thoracic ganglia of the sympathetic chain. We see that only $59.62 \pm 4.22\%$ of TH+ neurons are YFP+ and this explains the lack of decreased NE in the organs innervated by these ganglia. However, in SNS neurons in fat, the TH-Cre line proves to have complete co-localization of YFP and the marker TH (Zeng et al. 2015)."

This begs the question of which ganglia provide sympathetic innervation to fat, and what fraction of TH+ neurons in those ganglia are YFP+. Please specify, and show data from the relevant ganglia to prove adipose-specific effects.

A: It was already shown that the adipose tissue is innervated by neurons residing in the 13th ganglia of the sympathetic chain (Youngstrom, T.G. & Bartness, T.J. 1995).

To address the reviewer's question, we performed an immunostaining on the 13th ganglia of the sympathetic chain, which was meticulously microdissected according to methods described in Malin, S.A., Davis, B.M. & Molliver, D.C. 2007.

We now show in Supplementary Fig. 4c that $94.25 \pm 10.41\%$ of TH⁺ neurons colocalize with the conditional fluorescent reporter. This near complete co-localization is consistent with a preferential labeling of the TH-Cre driver for white adipose tissue.

In reference to your comment above, in which we “now assert that the BRAINSPAReDT technique (...) causes quite *specific* deletion of sympathetic neurons to adipose tissue, but not to other organs”. We would like to emphasize that, in our manuscript, we never stated that we developed a tool to *specifically* delete sympathetic neurons in adipose tissue, and that is not the main goal of this manuscript. We do claim a regional effect based on the *preferential* expression of the TH-Cre driver, which under-represents organs such as heart and lungs. Notwithstanding, we never stated that the TH-Cre driver is *specific* to adipose tissue, even though we see complete co-localization of YFP and the marker TH in fat, and in the 13th ganglia of the sympathetic chain. We have therefore used the word “*preferential*” in the text, rather than *specific*. Our goal at this point is solely to provide evidence that BRAINSPAReDT ablates sympathetic neurons outside the brain, which the reviewer did not oppose, rather than extending the scope of this work to the next level such as to prove organ *specificity*. This would certainly be exciting but it is a long-term endeavor that would require the development of additional tracing tools to disentangle which neurons innervate which fat pads across the body axis.